# Multiphase Model for Predicting the Thermal Conductivity of Cement Paste and Its Applications

**DOI:** 10.3390/ma14164525

**Published:** 2021-08-12

**Authors:** Yuanbo Du, Yong Ge

**Affiliations:** School of Transportation Science and Engineering, Harbin Institute of Technology, Harbin 150090, China; duyuan_bo@126.com

**Keywords:** cement paste, thermal conductivity, molecular dynamic method, curing temperature

## Abstract

Thermal conductivity plays a significant role in controlling thermal cracking of cement-based materials. In this study, the thermal conductivity of cement paste at an early age was measured by the hot plate method. The test results showed that the thermal conductivity of cement paste decreased with the increase of water/cement ratio and curing age. Meanwhile, a multiphase model for the thermal conductivity of cement paste was proposed and used to study the influence of saturation and curing temperature on the thermal conductivity of cement paste. To determine the parameters involved in this model, the thermal conductivity of each phase in cement paste was calculated by the molecular dynamic simulation method, and the hydration of cement was simulated by the Virtual Cement and Concrete Testing Laboratory. The inversion results showed that the relative error between experimental and simulation results lay between 1.1% and 6.5%. The thermal conductivity of paste in the saturated condition was 14.9–32.3% higher than that in the dry state. With the curing temperature increasing from 10 °C to 60 °C, the thermal conductivity of cement paste decreased by 3.9–4.9% depending on the water/cement ratio.

## 1. Introduction

Cement-based materials are the most widely used man-made materials in the world, but they are often threatened by an inevitable problem: early-age cracking [1,2]. Among many causes of early-age cracking in cement-based materials, thermal stress is the most important [3]. Cement-based materials are heterogeneous and are usually constituted by the cement paste and aggregate. The cement paste and aggregate have different thermal conductivity properties and result in thermal stress in the material, when temperature is generously changed. If thermal stress is beyond the shear stress of weak parts in cement-based material at an early age, it can lead to cracking. Such premature deterioration affects the mechanical properties and durability of a cement-based material [4,5,6]. Thus, a good knowledge of thermal conductivity of cement paste is essential in understanding the thermal cracking behavior of cement-based materials.

In the past few years, significant research has been carried out on the thermal conductivity of cement paste. According to previous studies [7,8,9], the water/cement ratio is a significant factor that affects the thermal conductivity of cement paste. In fact, the change in the water/cement ratio results in different porosity and free water content which leads to variations in thermal conductivity. The relationship between thermal conductivity of cement paste and the water/cement ratio has been reported to be linear and negative [9]. In addition, the moisture state also has an important effect on the thermal conductivity of cement paste, mainly because the thermal conductivity of water is about 25 times higher than that of air. It has been found that the thermal conductivity of cement paste in a saturated condition is 47% to 160% higher than that in dry conditions [10]. Moreover, the thermal conductivity of cement paste is also affected by supplementary cementitious materials due to the change of the microstructure of paste. For example, the addition of fly ash and slag has a decreasing effect on the thermal conductivity of the paste. However, the addition of silica fumes is in turn favorable to enhance the thermal conductivity of the paste [7,11]. A recent study revealed that the addition of pet coke also reduced the thermal conductivity of cement paste [12]. However, up to now, the thermal conductivity of constituent phases in the cement paste (such as AFt and AFm) is still unclear.

Currently, several prediction models for the thermal conductivity of cement paste have been established. Mounanga et al. [13] developed a numerical model to estimate the thermal conductivity of cement paste. The authors divided the hardened cement paste into five phases: anhydrous cement, hydration products, water, air and hydrated gel pores. The results showed that this model could be used to predict the thermal conductivity of cement paste at an early age. Moreover, Tang [7] developed a fractal model to determine the thermal conductivity of cement paste. In this work, cement paste was considered as a two–phase composite material consisting of solid phase and water (or air) phases. The predicted thermal conductivity of cement paste in a saturated state (or dry state) agreed well with the experimental data. Based on the proposed model, the influence of porosity on the thermal conductivity of cement paste was also interrogated. Kim et al. [9] studied the thermal conductivity of cement paste with different water/cement ratios, types of admixtures and humidity conditions. Based on the test results, a regression model for the thermal conductivity of cement paste was established. These models are helpful in understanding the thermal conductivity of cement paste. However, the research on the prediction model which can link constituent phase properties to the overall thermal conductivity of cement paste is still very minimal.

In view of the above, this paper studied the thermal conductivity of cement pastes with different water/cement ratios (0.34, 0.41 and 0.49) and curing ages (1, 3, 7, 14 and 28 days). Moreover, the thermal conductivity of each individual phase in paste was calculated by molecular dynamic simulation methods. The hydration process of cement was simulated by the Virtual Cement and Concrete Testing Laboratory (VCCTL). Based on those two studies, a multiphase model, which can link constituent phase conductivity to the overall thermal conductivity of cement paste, was proposed. In addition, the influence of saturation and curing temperature on the thermal conductivity of cement paste was studied with the proposed model.

## 2. Experimental Tests

### 2.1. Materials and Specimen Preparation

Type I Portland cement and tap water were used in this study. The chemical and mineralogical composition of Portland cement is shown in Table 1 and Table 2, respectively. The particle size distribution of cement is displayed in Figure 1. The water/cement (*W*/*C*) ratio of cement pastes was 0.34, 0.41 and 0.49, respectively. Firstly, the water and cement were mixed in a planetary-type mixer according to the Chinese standard GB/T 1346–2001 [14]. Then, fresh pastes were cast in the steel molds of 150 mm × 150 mm × 50 mm prisms, cured at room temperature and an ambient humidity for 24 h. After that, these samples were demolded and placed into a curing room (20 ± 1 °C, RH ≥ 98%) until the test age.

### 2.2. Thermal Conductivity Measurement

The thermal conductivity of cement pastes was tested by the hot plate method using a DRH-III thermal conductivity tester (Xiangtan instruments and meters Co., Ltd., Xiangtan, China). Prior to conducting the test, the specimens were stopped hydrating with ethanol and dried in an oven at 105 °C until their mass remained stable (±1 g). Then, the dried specimens were polished to achieve smooth surfaces and placed between the cold (15 °C) and hot (35 °C) plates to measure their thermal conductivity. Three specimens in each group were tested and the average was taken. The thermal conductivity of cement paste can be calculated by the formula as follows:(1)$$K=\frac{W\cdot d}{A\cdot (t_{1}-t_{2})},$$
where *K* is the thermal conductivity of the specimen (W/m K), *W* is the power of the heat-meter (W), *d* is the thickness of the specimen (m), *A* is the area of the specimen (m^2^), *t*_1_ is the temperature of the hot plate (K), and *t*_2_ is the temperature of the cold plate (K).

### 2.3. Experimental Results

Figure 2 shows the experimental results of thermal conductivity of cement pastes. It could be found that the thermal conductivity of cement paste decreased with increasing water/cement ratio. The thermal conductivities of air and water are 0.026 and 0.611 W/(m K), respectively. However, the thermal conductivity of the solid phase in cement paste is about 1.55 W/(m K) [15]. For cement paste with a higher water/cement ratio, a lower amount of solid phase exists in the cement paste, hence the paste exhibits a smaller thermal conductivity value. In addition, it can be seen from Figure 2 that the thermal conductivity of paste decreased with curing age. This is attributed to two effects. Firstly, with the increase of curing age, the porosity in paste reduces, the density of paste increases, and thus the thermal conductivity increases. Secondly, the thermal conductivity of clinker phases is higher than those of air phase and hydration products (see Section 4.1 for more details). The clinker phases in paste decrease with the increase of the curing age, resulting in a decrease in thermal conductivity. Based on the above, it could be found that the latter has a greater effect on the thermal conductivity of cement paste than the former.

## 3. Multiphase Model for the Thermal Conductivity of Cement Paste

Hardened cement paste is a composite material consisting of gas phase (air), liquid phase (water) and solid phase (hydration products and dehydrated cement). Therefore, a multiscale model was proposed to predict the thermal conductivity of cement paste in this study (see Figure 3). At the largest scale, the cement paste can be modeled by assuming the pores as inclusions and the rest phases as the matrix. At the next scale level, the matrix can be assumed to consist of the water phase as inclusions and the solid phase of the cement paste as the matrix. Again, the solid phase can be regarded as another two–phase composite with the cement clinker as inclusions and the hydration products as the matrix. At the lowest scale level, the cement clinker is modeled as a disordered assemblage of tricalcium silicate (C_3_S), dicalcium silicate (C_2_S), tricalcium aluminate (C_3_A), tetracalcium aluminoferrite (C_4_AF); meanwhile, the hydration products are considered as a disordered mix of calcium silicate hydrate (CSH), calcium hydroxide (CH), ettringite (AFT) and monosulphate (AFM). It should be noted that except for the above-mentioned phases, there are several other phases existing in cement paste, such as hydrogarnet (C_3_AH_6_), ironic hydroxide (FH_3_), calcium alumino–ferrite hydrate (C_4_(A,F)H_13_), etc. [16,17]. However, these phases are neglected in this model because the contents of these phases in cement paste are very low.

As shown in Figure 3, the Mori–Tanaka (MT) scheme and the Self-Consistent (SC) scheme were used to estimate the homogenized conductivity of the cement paste. The MT scheme was firstly proposed by Mori and Tanaka to study the elastic property of composites with a matrix-inclusion morphology [18]. The SC method was originally formulated by Hill (1965) for effective modulus of composites [19]. These schemes also can be used to predict the mechanical and thermal properties of cement-based materials [20]. Both models are based on the Eshelby solution. In the SC model, it is assumed that each inclusion is embedded in an infinite homogeneous matrix whose thermal conductivity is the same as that of the composite material. The SC estimation of the thermal conductivity for an N-phase composite is given by [21]:(2)$$K=\frac{\sum_{i=1}^{N}v_{i}k_{i}\frac{3K}{2K+k_{i}}}{\sum_{i=1}^{N}v_{i}\frac{3K}{2K+k_{i}}},$$
where *K* is the overall conductivity of the composite. *k**_i_* and *v**_i_* are the thermal conductivity and volume fraction of phase *i*, respectively.

According to the MT model, each inclusion behaves as an isolated inclusion in an infinite homogeneous matrix domain subjected to a constant far-field heat flux. For a composite material, the MT estimation of the effective thermal conductivity can be computed from [22]:(3)$$K=\frac{v_{m}k_{m}+\sum_{i=1}^{N}v_{i}k_{i}\frac{3k_{m}}{2k_{m}+k_{i}}}{v_{m}+\sum_{i=1}^{N}v_{i}\frac{3k_{m}}{2k_{m}+k_{i}}},$$
where *K* is the thermal conductivity of the composite material, *k**_m_* and *v**_m_* are the thermal conductivity and volume fraction of the matrix, respectively*,* and *k**_i_* and *v**_i_* are the thermal conductivity and volume fractions of the *i*th inclusion, respectively. *N* is the number of inclusion phases.

## 4. Model Parameters

To use the models described in Section 3 to calculate the thermal conductivity of cement paste, it is necessary to know the thermal conductivity and volume fraction of each phase in the cement paste. These parameters will be discussed in this section.

### 4.1. Thermal Conductivity of Constituent Phases in Cement Paste

Qomi, Ulm and Pellenq [23] have predicted the thermal conductivity of C_3_S, C_2_S, CSH and CH by molecular dynamic (MD) simulation and demonstrated that thermal conductivities of C_3_S, C_2_S, CSH and CH were 3.35, 3.45, 0.98 and 1.32 W/(m K), respectively. The thermal conductivity of air is 0.026 W/(m K). However, the thermal conductivity of other phases in the cement paste (such as C_3_A) is still unknown. Note that experiments for studying the thermal conductivity of the constituent phase in cement paste is difficult. For example, it is difficult to obtain a high purity of constituent phase in the laboratory (when we use CaCO_3_ and Al_2_O_3_ to produce C_3_A, there are always some free CaO in C_3_A). The MD simulation provides a convenient and economic way to study and estimate the thermal conductivity of materials. Previous research [24,25] has demonstrated that the simulated result by the MD method is in qualitative agreement with the measured data. Thus, the MD method was adopted here to investigate the thermal conductivity of each phase in cement paste.

#### 4.1.1. Theory Background

In MD, the thermal conductivity of materials can be computed by two generic approaches including Equilibrium Molecular Dynamics (EMD) [26] and Non-Equilibrium Molecular Dynamics (NEMD) [27]. The NEMD method is based on Fourier’s law, and the temperature gradient of the simulation cell is used to calculate thermal conductivity of a material. In contrast, the EMD method is developed according to the Green–Kubo formula, and the integral of the heat current autocorrelation function of the simulation cell is used to calculate thermal conductivity of a material. In general, the EMD method possesses the features of lower size effect and higher stability [28]. Thus, the EMD method was adopted in this study. According to the Green–Kubo theory, the thermal conductivity of atomic structure can be calculated by the following equation [29]:(4)$$k=\frac{1}{3k_{B}VT^{2}}\int_{0}^{\infty }<J(0)J(t)>dt,$$
where *k* is the thermal conductivity. *k_B_* is the Boltzman constant (1.38 × 10^−23^ J/K). *T* is the absolute temperature (K). *V* is the volume. < > denotes the average over different time origins. <**J**(0)**J**(*t*)> is the heat current autocorrelation function (HCACF).

Since the MD simulation is performed for discrete time steps. To use the above equation in MD simulation, the integration of the equation should be represented in the discrete form as:(5)$$k(t_{m})=\frac{\Delta t}{3k_{B}VT^{2}}\sum_{m=1}^{M}\frac{1}{N-m}\sum_{n=1}^{N-m}\left(J(m+n)J(n)\right),$$
where ∆*t* is the simulation time step. *N* represents the total number of time steps of the simulation (after equilibrium). *M* represents the calculation steps used for calculating average value. The outer summation in Equation (5) represents the HCACF. Details of the procedure of the simulation will be discussed in Section 4.1.2.

#### 4.1.2. Simulation Details and Results

The MD simulations were carried out by using the large-scale atomic/molecular massively parallel simulator (LAMMPS). In MD simulation, the 2 × 2 × 2 supercells of the clinkers and hydration products [30,31,32,33,34,35] were used (Figure 4). The ClayFF force field (C_3_A and C_4_AF) and CVFF force field (C_3_S, C_2_S, AFt, AFm) were applied in simulations [36]. All MD simulations were performed with the periodic boundary condition. The time step was 1 fs. First, the initial condition of the system is obtained by a random number at 298 K. Then, the system is equilibrated in the isobaric–isothermal ensemble (NPT, 298 K, 0 atm) for 0.5 ns. The environmental temperature and pressure are controlled by using the Nose–Hoover thermostat. Thereafter, the system is switched to microcanonical ensemble (NVE) evolving for 1000 ps (or 20 ps). During this stage, the heat current data were recorded every 10 steps. Finally, the thermal conductivity was calculated according to Green–Kubo theory with the help of Equation (5). Fifteen independent runs were performed to minimize the uncertainty. The running thermal conductivities for the clinkers and hydration products were shown in Figure 5.

From Figure 5 it could be found that the simulated thermal conductivity for C_3_S and C_2_S were, respectively, 3.35 ± 0.3 and 3.45 ± 0.2 W/(m K), which are in good agreement with the recent simulated results of Qomi et al. [23]. This indicated that the simulation model and corresponding parameters used in this study were reasonable. The simulated thermal conductivity for C_3_A was 3.74 ± 0.2 W/(m K), while the simulated thermal conductivity for C_4_AF was 3.17 ± 0.2 W/(m K). In addition, it could be seen from Figure 5 that the simulated thermal conductivity for AFT and AFM were, respectively, 2.19 ± 0.4 and 2.56 ± 0.2 W/(m K), which are both smaller than those of clinker phases.

### 4.2. Volume Fraction of Constituent Phase in Cement Paste

In the early stage of hydration, cement paste develops from a quasi-fluid state to a solid state, and the volume fractions of constituent phases in paste change sharply. In order to simulate the hydration process of cement paste, several models have been developed, such as the Jennings–Tennis model [37], CEMHYD3D model [38], VCCTL model [39] and HYMOSTRUC model [40]. Among them, the VCCTL model is an extension of the EMHYD3D modeling package. Previous research [41,42] has proven that the VCCTL model could be successfully used to reproduce the hydration process of cement paste. Therefore, the VCCTL model was used in this study to simulate the hydration of cement and obtain the volume fraction of constituent phases in the paste. The mixture proportions and material properties used in VCCTL model were described in Section 2. Isothermal conditions of 20 °C was used for all mixes and the time conversion factor of β was 2.4 × 10^−4^ h/cycle^2^ [43]. The simulated development of the volume fraction of each phase in cement paste (*W*/*C* = 0.34, 0.41, and 0.49) is shown in Figure 6. As expected, the volume fractions of pore and clinker phases in all three pastes decreased with curing age, while the volume fractions of hydration products increased with curing age. In addition, this trend was more obvious during the first seven days, which reflected the higher rates of hydration at the early age.

## 5. Model Validation and Application

### 5.1. Comparison of Experimental Data with Model Predictions

Based on the model described in Section 3 and the characteristic parameters of each phase in cement paste covered in Section 4, the thermal conductivity of cement pastes was calculated and compared with the experimental results, as shown in Figure 7. It could be found that the predicted thermal conductivity values of cement paste using the above described model were by and large consistent with those of experimental measurements. The relative error between experimental and simulation results was between 1.1% and 6.5%. Khan [44] used the Harmathy’s model to predict the thermal conductivity of cement-based materials in saturated conditions. The result showed that the calculated values deviated from the measured values by 17% to 49%. Honorio et al. [45] proposed a multiscale model for the thermal conductivity of cement paste. In this work, the percentage errors between the experimental and predicted values are 7%, 11%, 14% and 15% for *W*/*C* = 0.40, 0.35, 0.30 and 0.25. Liu et al. [46] developed a multi-scale micromechanical model to determine the thermal conductivity of cement-based materials. The relative errors between the predicted and measured values are in the range of 0.5% to 22.2%. These results imply that the proposed model in this study is able to reliably predict the thermal conductivity of cement paste. It should be noted that the model described in Section 3 is established for pure cement paste. Therefore, the proposed model cannot be used to predict the thermal conductivity of cement paste with mineral admixtures (fly ash, slag, silica fume, etc.).

### 5.2. Influence of Saturation and Curing Temperature on the Thermal Conductivity of Paste

Curing temperature and saturation have a considerable effect on the composition and microstructure of cement paste. The proposed model validated in the previous section is exploited here to study the influence of saturation (0 to 1) and curing temperature (10 °C to 60 °C) on the thermal conductivity of paste (28 days). The results are shown in Figure 8 and Figure 9.

From Figure 8, it could be found that the thermal conductivity of cement paste increased approximately linearly with an increasing degree of saturation. From the dry to saturated state, the thermal conductivity of cement paste with *W*/*C* of 0.34, 0.41 and 0.49 increased by 14.9%, 23.0% and 32.3%, respectively. The increase in the thermal conductivity is attributed to replacement water with air in the pores of paste, since water has a much higher thermal conductivity (0.611 W/(m K)) than air (0.026 W/(m K)). Moreover, it could be observed from the figures that the higher the water/cement ratio of the paste specimen, the more rapidly the thermal conductivity increased. This is due to the influence of porosity.

As shown in Figure 9, the thermal conductivity of cement paste decreased with the increasing curing temperature. When the curing temperature increased from 10 °C to 60 °C, the thermal conductivity of cement paste decreased only by 3.9% (*W*/*C* = 0.34), 4.4% (*W*/*C* = 0.41) and 4.9% (*W*/*C* = 0.49), respectively. This indicated that the change of curing temperature had little influence on the thermal conductivity of cement paste. The increase of the curing temperature can accelerate the hydration of the cement clinker, resulting in the decrease of the volume fraction of clinker phases in the paste specimen. Because the clinker phases have higher thermal conductivity than hydration products and the water and air phase, decreasing their volume fraction causes a logical decrease of the thermal conductivity of paste. The consequence of increasing the curing temperature is indeed similar to the increasing the hydration time.

## 6. Conclusions

In this study, the thermal conductivity of cement pastes was measured and predicted by a newly proposed multiphase model. The VCCTL software and MD simulation method were adopted to calculate the character parameters of constituent phases in paste. Based on the experiments and calculations, the following conclusions can be drawn.

The thermal conductivity of cement paste decreased gradually with the extension of curing age. For the same curing age, the thermal conductivity of cement paste decreased with the increase of the water/cement ratio.By combining the Mori–Tanaka method and Self-Consistent method, a multiscale prediction model for the thermal conductivity of cement paste was proposed, which can link constituent phase conductivity to the overall thermal conductivity of cement paste. The relatively errors between the predicted and measured values are in the range of 1.1% to 6.5%.The simulated thermal conductivity for C_3_S and C_2_S were, respectively, 3.35 ± 0.3 and 3.45 ± 0.2 W/(m K), which agreed well with the previously published results. In addition, the simulated thermal conductivities for C_3_A, C_4_AF, AFT and AFM were 3.74 ± 0.2, 3.17 ± 0.2, 2.19 ± 0.4 and 2.56 ± 0.2 W/(m K), respectively.The thermal conductivity of cement paste increased linearly with an increasing degree of saturation. The thermal conductivity of cement paste in saturated conditions was 14.9%–32.3% higher than that in a dry state. Moreover, the increase of curing temperatures from 10 °C to 60 °C resulted in a 3.9%–4.9% decrease in thermal conductivity of cement paste.

## Figures and Tables

**Figure 1 materials-14-04525-f001:**
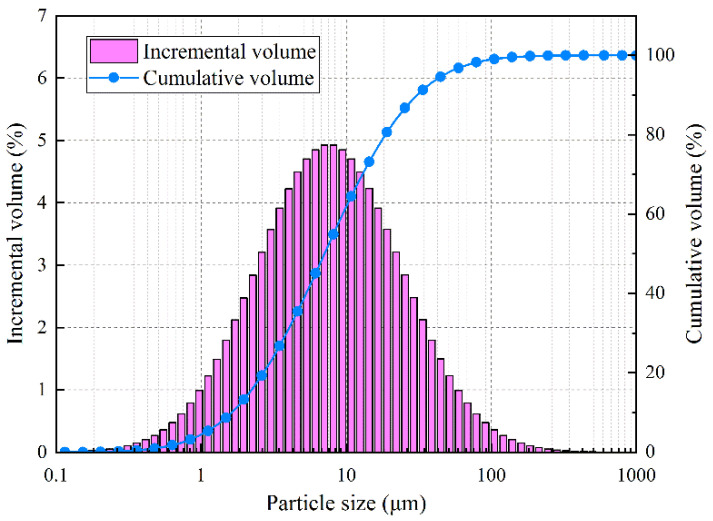
Particle size distribution of cement.

**Figure 2 materials-14-04525-f002:**
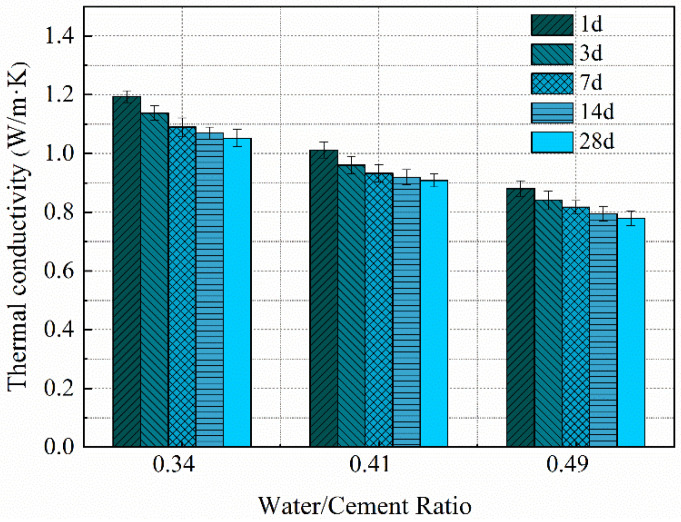
Thermal conductivity of cement pastes with different water/cement ratios.

**Figure 3 materials-14-04525-f003:**
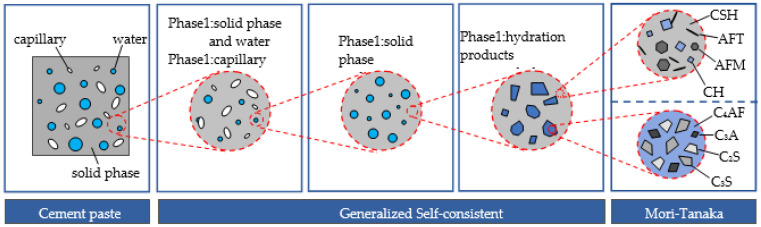
Multiphase modeling of cement paste.

**Figure 4 materials-14-04525-f004:**
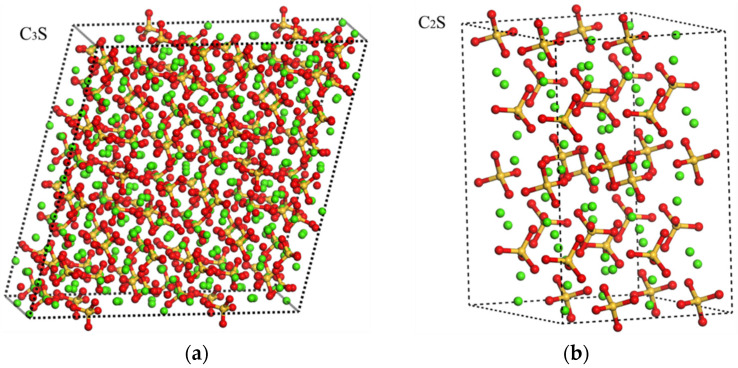
A 2 × 2 × 2 supercell molecular structure of crystal minerals (the white, red, blue, yellow, pink, grey and green balls represent the H, O, S, Si, Al, Fe and Ca atoms, respectively): (**a**) C_3_S, (**b**) C_2_S, (**c**) C_3_A, (**d**) C_4_AF, (**e**) AFt, (**f**) AFm.

**Figure 5 materials-14-04525-f005:**
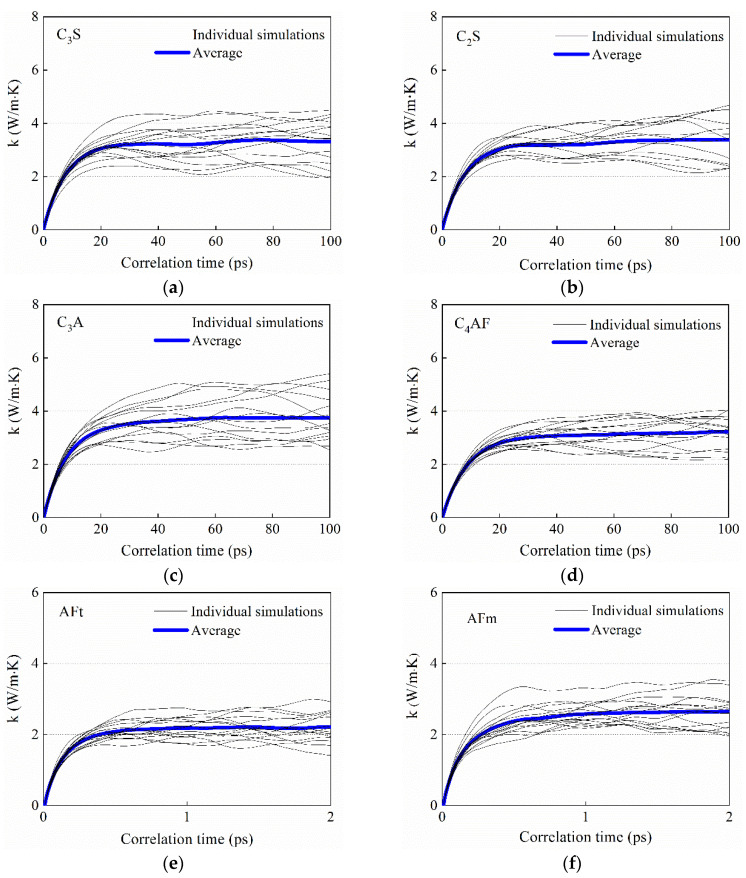
Running thermal conductivities as a function of correlation time for clinker phases and hydration products (the grey lines represent the independent simulation results, the blue lines represent the average value of 15 independent simulations): (**a**) C_3_S, (**b**) C_2_S, (**c**) C_3_A, (**d**) C_4_AF, (**e**) AFt, (**f**) AFm.

**Figure 6 materials-14-04525-f006:**
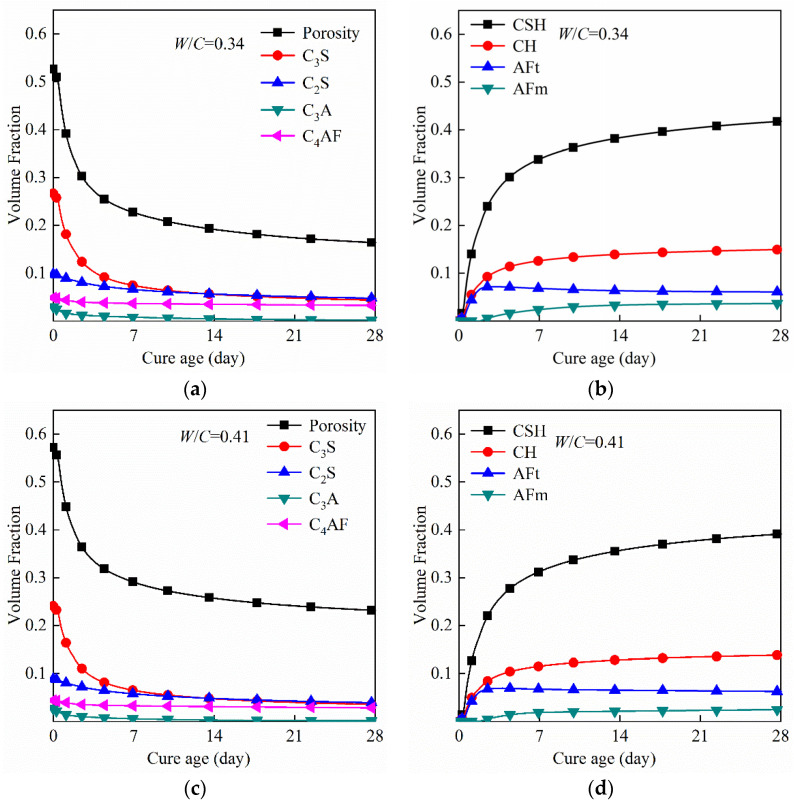
The simulated development of volume fraction of each phase in the cement paste during the hydration period: (**a**,**b**) *W*/*C* = 0.34, (**c**,**d**) *W*/*C* = 0.41, (**e**,**f**) *W*/*C* = 0.49.

**Figure 7 materials-14-04525-f007:**
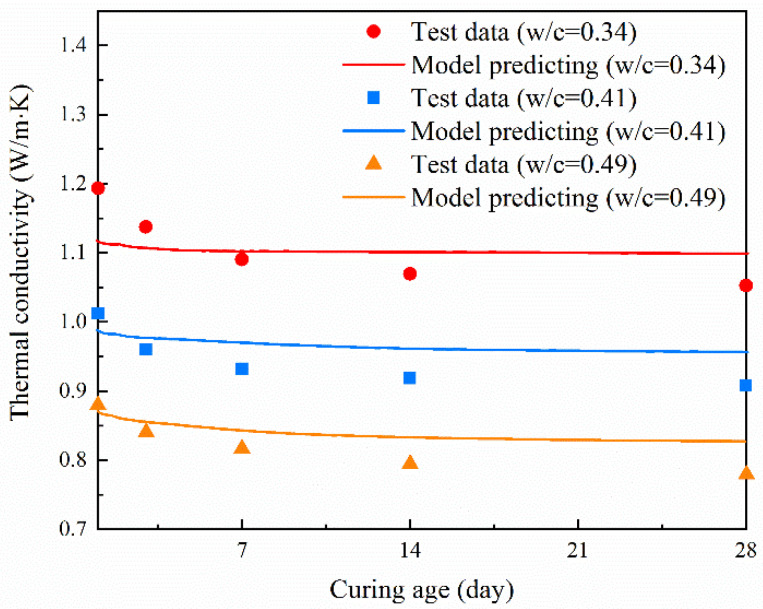
Thermal conductivity of cement pastes: estimations compared to experimental data.

**Figure 8 materials-14-04525-f008:**
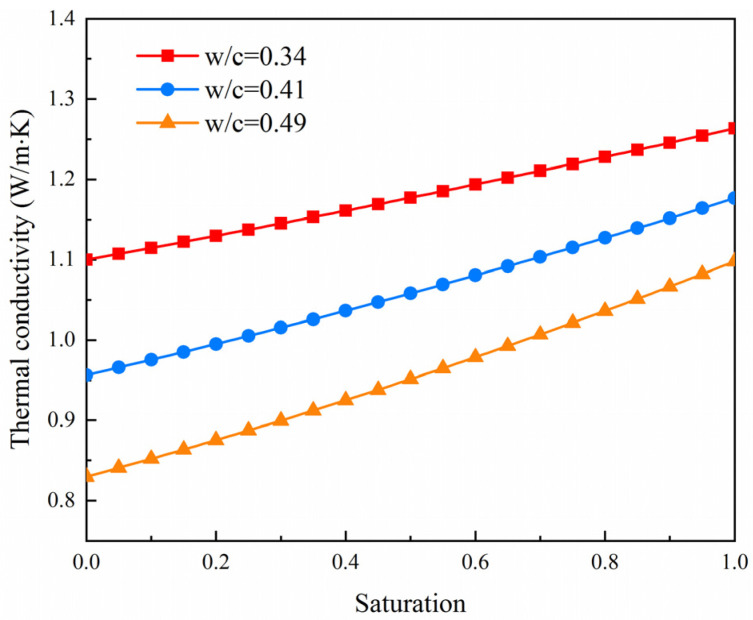
Evolution of the thermal conductivity against curing temperature at dry state.

**Figure 9 materials-14-04525-f009:**
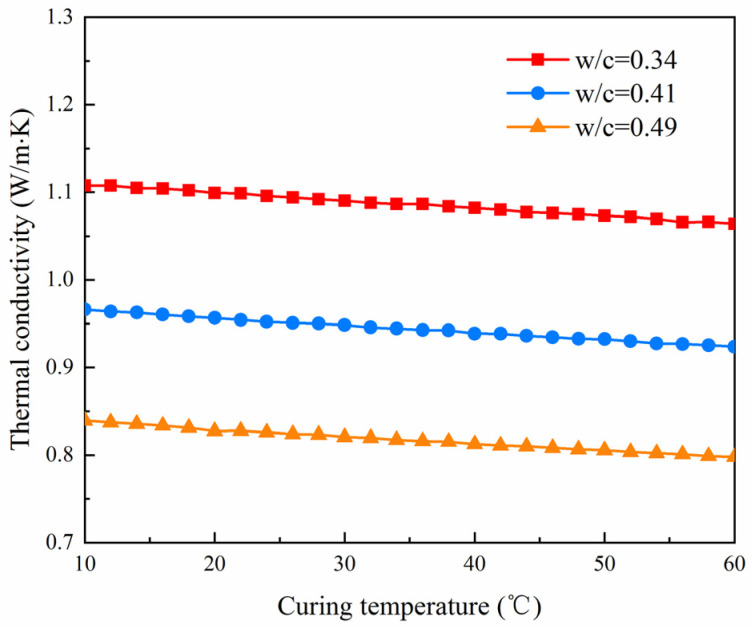
Evolution of the thermal conductivity against curing temperature at dry state.

**Table 1 materials-14-04525-t001:** Chemical compositions of cement.

Components	Al_2_O_3_	SiO_2_	Fe_2_O_3_	CaO	MgO	Na_2_O	SO_3_	Loss of Ignition
Contents/wt%	4.49	21.88	3.45	64.65	2.36	0.51	2.44	1.31

**Table 2 materials-14-04525-t002:** Mineralogical composition of cement.

Phase	C_3_S	C_2_S	C_3_A	C_4_AF	CaSO_4_·2H_2_O
Contents/wt%	56.54	20.87	6.22	10.31	6.06

## Data Availability

The data presented in this study are available on request from the corresponding author.

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
