# Peer review of "Multiphase Model for Predicting the Thermal Conductivity of Cement Paste and Its Applications"

_materials, 2021, doi:10.3390/ma14164525_

Round 1

Reviewer 1 Report

A careful review of the manuscript “Linking constituent phase properties to thermal conductivity of cement paste: A new prediction model and its applications” has been completed. Even though the authors tried to evaluate the thermal conductivity of cement paste, it is not clear what is the main objective of the manuscript and how these evaluations can be used in practice; also, it is very difficult to follow what is presented. This paper is useful for engineers to evaluate the thermal conductivity of paste. However, this investigation is not comprehensive and there are still rooms to improve. Therefore, this manuscript is not recommended for publication in International Journal of materials since paper has a critical and serious problem explained below:

  1. English needs to be improved. I had difficulty to follow the text and had to read the same sentence several times.
  2. The originality is not explained in detail.
  3. References are not cited sufficiently and appropriately.

Reviewer 2 Report

Thank you for giving me the qualifications to review this paper. I checked the whole thing. There are a few more things to consider in order to increase the effectiveness of the simulation. Please confirm. 

Minor

The captions of the 6 graphs in Figure 6 should be organized.
The 3 graphs on the right and 3 graphs on the left must each have the same Axis scale.
Absolute comparison is not possible.
It is recommended to change from a 2X3 layout to a 3X2 layout.

Major

The data the authors explain the results of figure 5 are strange. On the graph, in the case of C3A, the deviation seems to be more than 1W/mK but author write 3.74±0.2. And if the deviation is this large, I think there is a limit to applying it to actual modeling.

The author explains that the thermal conductivity of cement is predictable based on the results of Figure 7. However, the actual simulation results show that the results are reversed according to the curing age, while at the same time the difference is widening over time. Over time, these differences may widen. It is judged that the conclusion was reached too quickly in a situation where the difference between the measurement result of the actual material and the model result would increase further.

Also, as a result of the curing process, the content of water will continue to decrease, so isn't it necessary to consider this more? The structure of the void will change during the drying process depending on the water content, and it is necessary to interpret this, but it is judged that the data is insufficient.

Data were obtained using saturation and temperature conditions for model validation. What does this mean for modeling predition? Shouldn't the result be reconnected with the previous figure 7, resulting in a modified model expression? 

Reviewer 3 Report

REVIEW

on article

Linking constituent phase properties to the thermal conductivity of cement paste: A new prediction model and its applications

Yuanbo Du and Yong Ge

SUMMARY.

The article is devoted to the development of a model for predicting the thermal conductivity of cement paste with different water/cement ratios and curing ages. In general, the prediction of the thermal conductivity of cement is an important scientific problem of energy saving in buildings and creating a comfortable indoor environment.

The authors consider cement stone as a multi-phase multi-level medium in which the matrix consists of an aqueous phase in the form of inclusions and a solid phase. The authors also consider the solid phase as another two-phase composite with cement clinker as inclusions and hydration products as a matrix. To assess the homogenized conductivity of the cement paste, the Mori-Tanaka scheme and the self-consistent scheme were used.

The reference list contains 49 items.

However, there are shortcomings and ambiguities that need to be corrected.

COMMENTS.

  1. The authors have to redo the Abstract and bring it in line with the requirements of the Materials journal. The scientific novelty is poorly defined. The authors need to highlight it. Editors strongly encourage authors to use the following style of structured abstracts, but without headings: (1) Background: Place the question addressed in a broad context and highlight the purpose of the study; (2) Methods: Describe briefly the main methods or treatments applied; (3) Results: Summarize the article's main findings; and (4) Conclusions: Indicate the main conclusions or interpretations. The abstract should be an objective representation of the article.
  2. The title, it seems to me, does not correspond to the essence of the article. I would suggest changing to "Multiphase model for predicting the thermal conductivity of cement paste and its applications." But this is only a recommendation, not a remark. The decision is up to the authors.
  3. “Abstract: The thermal conductivity of cement-based materials plays a significant role in controlling the amount of heat transfer through the buildings.” This is a banal phrase; it needs to be removed and the scientific significance of the problem is emphasized.
  4. Section 4. The problem statement is not clear. The authors need to formulate the problem as a problem of thermal conductivity.
  5. Describe the details of the simulation in more detail. What algorithms were used? What initial data were used?
  6. What are the boundary conditions of the problem?
  7. I don't understand what Correlation Time is. Maybe just Time?
  8. Section 5. Where did the authors get the experimental results? Describe in more detail the experimental conditions and the method for determining the thermal conductivity at the non-molecular level.
  9. In the Discussion section, the authors need to compare the results obtained with those of other researchers. It is desirable to show the limits of applicability of the model.
  10. The conclusions need to be redone in accordance with the corrections in the text of the article.

In general, the article is devoted to an interesting and important scientific problem that will undoubtedly attract the attention of readers.

However, there are many fixes. I recommend the article for publication after a major correction.

Reviewer 4 Report

This is an interesting topic and also very important. However, there are some issues which need to be cleared before the paper can be accepted for publication:

1) major issue: how can we state the thermal conductivity at 1, 3, 7... days, when we first dry the specimens at 105 deg. C. This is effecting the curing age considerably. Please commnet and give an explanation.

2) Equations (2) and (3) are given without explanation or background (they were cited, but at least a sentence on what is the background would be approspriate).

3) In sectoins 4.1.1 and 4.1.2 a simulations procedure is shown. However, the explanation and details are not given in order for someone to repeat the research. Why are there the differences in simulations, what parameters are random and what are the distributions?

4) If there is a scatter of fraction Ki how come the overall agreement between the experiment and simulation is below 7 %. What was the scatter in experiments. How many repetitions were performed.

5) In the analyses of curing temeprature influence there is one very important questions which was not addressed: higher temperature has a strong influence on hydration rate and thus on curing age. Please, explain.

There are some minor comments in the attached paper.

Round 2

Reviewer 1 Report

Very similar issues still showing!

Reviewer 3 Report

All my remarks were taken into account and the text of the article has been corrected. I recommend the article for publication.